# Interactome of the HIV-1 proteome and human host RNA

Tinus Schynkel[1], Willem van Snippenberg [1,2], Kimberly Verniers[2], Gwendolyn M Jang[3,4,5], Nevan J Krogan [3,4,5], Pieter Mestdagh[2], Linos Vandekerckhove [1,6✉] & Wim Trypsteen [1,2,6✉]

## Abstract

The human immunodeficiency virus (HIV-1) is highly dependent on a variety of host factors. Beside proteins, host RNA molecules are reported to aid HIV-1 replication and latency maintenance. Here, we implement multiple workflows of native RNA immunoprecipitation and sequencing (nRIPseq) to determine direct host RNA interaction partners of all 18 HIV-1 (poly)proteins. We identify 1,727 HIV-1 protein – human RNA interactions in the Jurkat cell line and 1,558 interactions in SupT1 cells for a subset of proteins, and discover distinct cellular pathways that seem to be used or controlled by HIV-1 on the RNA level: Tat binds mRNAs of proteins involved in the super elongation complex (AFF1-4, Cyclin-T1). Correlation of the interaction scores (based on binding abundancy) allows identifying the highest confidence interactions, for which we perform a small-scale knockdown screen that leads to the identification of three HIV-1 protein binding RNA interactors involved in HIV-1 replication (AFF2, H4C9 and RPLP0).

Keywords HIV-1; RIPseq; Interactome; Protein Interactions; Host RNA
Subject Categories Microbiology, Virology & Host Pathogen Interaction; RNA Biology

## Introduction

The human immunodeficiency virus (HIV-1) is a relatively compact biological entity: nine genes encoding for 15 proteins. Remarkably, with only these tools available, HIV-1 is able to infect CD4+ cells, replicate, and perform immune evasion. Moreover, although combinational antiretroviral therapy (cART) can control an HIV-1 infection, it cannot eliminate the virus, as HIV-1 can initiate and maintain a latent state that causes viral rebound upon treatment cessation (Siliciano and Greene 2011). To achieve all this, HIV-1 is heavily dependent on host factors, ranging from co-receptors for viral entry to the ribosomal translation machinery of the host cell (Brass et al, 2008; Zhou et al, 2008; König et al, 2008). Historically, research was mainly focused on examining direct protein-protein interactions (PPIs) to discover host proteins, complexes, and pathways hijacked by the virus, while other types of host dependency or restriction factors (HDFs/RFs) were somewhat overlooked (Jäger et al, 2011). However, in the last decade, it has become clear that there are other major RNA classes exerting cellular functions (ENCODE Project Consortium, 2012). Increasing numbers of non-coding genes, including small nuclear RNAs, small nucleolar RNAs, microRNAs and long non-coding RNAs (lncRNAs) are being discovered and functionally characterized (Statello et al, 2021). Especially, lncRNAs are an interesting and emerging class of regulatory molecules that have unique features: transcription at lower levels, a poorly conserved sequence across species, but more cell-type and disease-state-specific expression (Statello et al, 2021; Iyer et al, 2015). To date, several lncRNAs, including the non-coding repressor of NFAT (NRON) (Li et al, 2016) and LINC01426 (Huan et al, 2018), have been reported to play a role in HIV-1 infectivity and/or latency maintenance by directly interacting with HIV-1 proteins. Besides non-coding RNAs, also host messenger RNA (mRNA) binding with HIV-1 proteins has proven to perform regulatory roles in the HIV-1 replication cycle (Kutluay et al, 2014; Bieniasz and Telesnitsky 2018). Thus, to understand the complete virus-host interplay, it is crucial to uncover the RNA interaction partners of all HIV-1 proteins. However, no large-scale efforts via HIV-1-host RNA interactor profiling have been reported yet.

Here, we performed native RNA immunoprecipitation and sequencing (nRIPseq) in the T lymphocyte-based Jurkat cell line for each of the 15 HIV-1 proteins, as well as for its three primary polyprotein products (Gag, Pol, Env) to construct a full HIV-1 protein–host RNA interactome. Additionally, for (poly)proteins showing a high RNA-binding capacity, nRIPseq was performed using alternative antibodies and in a second cell line (SupT1), strengthening this extensive HIV-1 protein–host RNA interactome and uncovering a variety of pathways involved in the HIV-1 replication cycle at the RNA level. Finally, an antisense oligonucleotide (ASO) knockdown of the most abundant RNA interactors in SupT1 cells revealed three RNA interactors (AFF2, H4C9, and RPLP0) for which knockdown inhibited HIV-1 replication.

[1]HIV Cure Research Center, Department of Internal Medicine and Pediatrics, Ghent University and Ghent University Hospital, Ghent 9000, Belgium. [2]OncoRNALab, Center for Medical Genetics (CMGG), Ghent University, Ghent 9000, Belgium. [3]Department of Cellular and Molecular Pharmacology, University of California, San Francisco, CA 94158, USA. [4]Quantitative Biosciences Institute (QBI), University of California, San Francisco, CA 94158, USA. [5]J. David Gladstone Institutes, San Francisco, CA 94158, USA. [6]These authors contributed equally: Linos Vandekerckhove, Wim Trypsteen. ✉E-mail: Linos.Vandekerckhove@UGent.be; Wim.Trypsteen@UGent.be

# Results

## Native RIPseq reveals host RNA interaction partners of HIV-1 proteins

To unravel the interactome of HIV-1 proteins and human RNAs, native RNA immunoprecipitation with subsequent RNA sequencing (nRIPseq) was performed on 18 Jurkat cell lines that each express an individual double-tagged HIV-1 (poly-)protein (2xStrepTagII-TEV-3xFLAG) upon doxycycline induction (Fig. 1A,B—assay A). To increase the enrichment of true RNA interactors and reduce aspecific signals, two nRIPseq background controls were included (nRIPseq of tagged GFP and nRIPseq with a random mouse IgG antibody), and for each HIV-1 protein nRIPseq experiment, the optimal signal-to-noise ratio was determined (Appendix Table S1). Western blot was performed to confirm the specific pulldown of the HIV-1 protein (Appendix Fig. S1). This full interactome revealed a total set of 1467 HIV-1 protein–host RNA interactions, with Nucleocapsid being the HIV-1 protein that possesses the highest host RNA binding capacity with 857 interactions, followed by Rev (289), Pol (73) and Gag (73) (Fig. 1C). The other HIV-1 proteins displayed a more limited host RNA binding profile with less than 50 interactions. Reverse transcriptase (RT) was the sole HIV-1 protein for which no host RNA interactions were identified.

Next, we aimed to validate this HIV-1 protein–host RNA dataset by performing three complementary nRIPseq workflows on a focused set of HIV-1 proteins (Fig. 1B). First, Strep-based pulldowns were performed for six of the HIV-1 proteins, displaying the highest RNA binding capacity in Jurkat cell lines (Nucleocapsid, Rev, Pol, Gag, Matrix, Tat) (Assay B; Appendix Fig. S2). Second, for three HIV-1 proteins (Nucleocapsid, Matrix, Tat) demonstrating the strongest host RNA interactions, we created a second cell line of interest (SupT1 cells) and performed FLAG-based nRIPseq, both in the presence and absence of an HIV-1 infection (Assays C and D; Appendix Figs. S3, S4). All detected interactions, as well as the count tables of all nRIPseq experiments, can be found as supplemental datasets (Datasets EV1, EV2).

In general, 86.1% of the identified RNA interactors of HIV proteins were protein-coding mRNAs, 3.0% lncRNAs, 6.8% pseudogenes, 2.3% tRNAs, and 1.5% non-coding small RNAs (Fig. 1D; Appendix Table S2), indicating a wide variety of RNA families that are recruited during HIV-1 replication. Using Strep-based nRIPseq, the majority of the detected RNA interactors of both Gag (95%) and Rev (65%) were tRNAs.

The four workflows of the nRIPseq platform displayed a significant overlap (Fisher exact tests in Appendix Table S3) in detected RNA interaction partners: for Nucleocapsid, 40.7% of the RNA interactors were detected by at least 2 out of 4 nRIPseq workflows (Fig. 2A). For Matrix and Tat this was 8.7 and 10.7% respectively, but the fact that one workflow detected a considerable higher number of RNA interactors for those proteins, lowered these percentages substantially. To be able to compare detected interactions between proteins and workflows, we developed an interaction score based on the reads per million (RPM) difference of an RNA interaction partner with the RPM of the background controls over all three replicates. This revealed a significant correlation between interaction scores for all four nRIPseq workflows (Fig. 2B, Appendix Fig. S6; Appendix Table S5). Only for the

proteins Tat and Matrix, assay B (Strep-based nRIPseq in Jurkat cells) did not correlate with assay C and D (in SupT1 cells). This could partially be explained by the potential masking of the RNA binding place of the protein by the alternative pulldown method. When comparing proteins, the top RNA interaction partners of Matrix have by far the highest interaction scores, up to 50 times higher than the average top RNA interaction partners (Fig. 2C), indicating strong and abundant interactions that are specific (as not more than 50 interactions were detected, each with a strong correlation between assay A, C, and D—Figs. 1C,2B). Other proteins displaying strong, abundant RNA interactions are Nucleocapsid, Tat, Integrase, Gag, and Rev. The few interactions found for GP41, Vpr, and Vif are rather weak and sparse (Fig. 2C).

Interestingly, besides a considerable overlap (Appendix Table S5) between the nRIPseq results in uninfected SupT1 (assay C) and HIV-1 infected SupT1 cells (assay D), we can also distinguish distinct RNA interactions detected in only one of both conditions. For Tat, 18 interactors are identified by both assays, but the presence of an HIV-1 infection provoked an additional 140 detected Tat–host RNA interactions. Similarly, for Nucleocapsid and Matrix we observed differences in host RNA interactions based on the infection status, suggesting that the cellular environmental context influences the RNA binding preferences of the HIV-1 protein (Fig. 2A). All RNA interactors identified in at least two nRIPseq workflows (Fig. 2A) as well as the top 10 RNA interactors for each HIV-1 protein and nRIPseq workflow (Fig. 2C), can be found as a browsable table in supplemental Dataset EV2.

Next, we constructed an interactome network view of the 1727 HIV–human interactions identified in the Jurkat cell line containing nodes corresponding to 18 HIV poly(proteins) (blue) and 1393 human RNAs (Fig. 3). The network representation of the 1558 interactions identified in the SupT1 cell line for Nucleocapsid, Matrix, and Tat can be found in supplemental (Appendix Fig. S7).

## Analyzing the HIV-1 protein–host RNA interactome(s): Gene ontology and motif enrichment

The interaction network, combined with gene ontology (GO) enrichment analysis performed on the set of interacting RNAs for each HIV-1 protein, revealed a variety of RNA clusters for which the protein products are involved in specific biological processes (Fig. 4A; Appendix Table S6). One remarkable observation is the fact that Tat interacts on an RNA level with a number of genes that are part of the ELL-containing super elongation complex (GO:0032783): AFF1, AFF2, AFF3, AFF4, CCNT1, and 7SK RNA. 7SK RNA is an abundant small nuclear RNA that sequesters the positive transcription elongation factor, P-TEFb, a complex that is recruited by Tat for the phosphorylation of paused RNA polymerase II and re-initiation of transcription (Bartholomeeusen et al, 2012; Peterlin et al, 2012; Sedore et al, 2007). AFF1 and AFF4 have been shown to increase the affinity of Tat for P-TEFb subunit cyclin-T1 (CCNT1) to facilitate P-TEFb extraction from 7SK RNA (Lu et al, 2014; Gu et al, 2014). One hypothesis is that Tat, in complex with P-TEFb, binds the mRNAs of these factors in a feed-forward loop to enhance their transcription and HIV-1 transcription elongation. To further validate these Tat interactors identified by our nRIPseq workflow, we performed a limited set of qPCR validations to assess enrichment levels in a targeted fashion. Interestingly, we could confirm 16.1- and 4.1-fold enrichment of

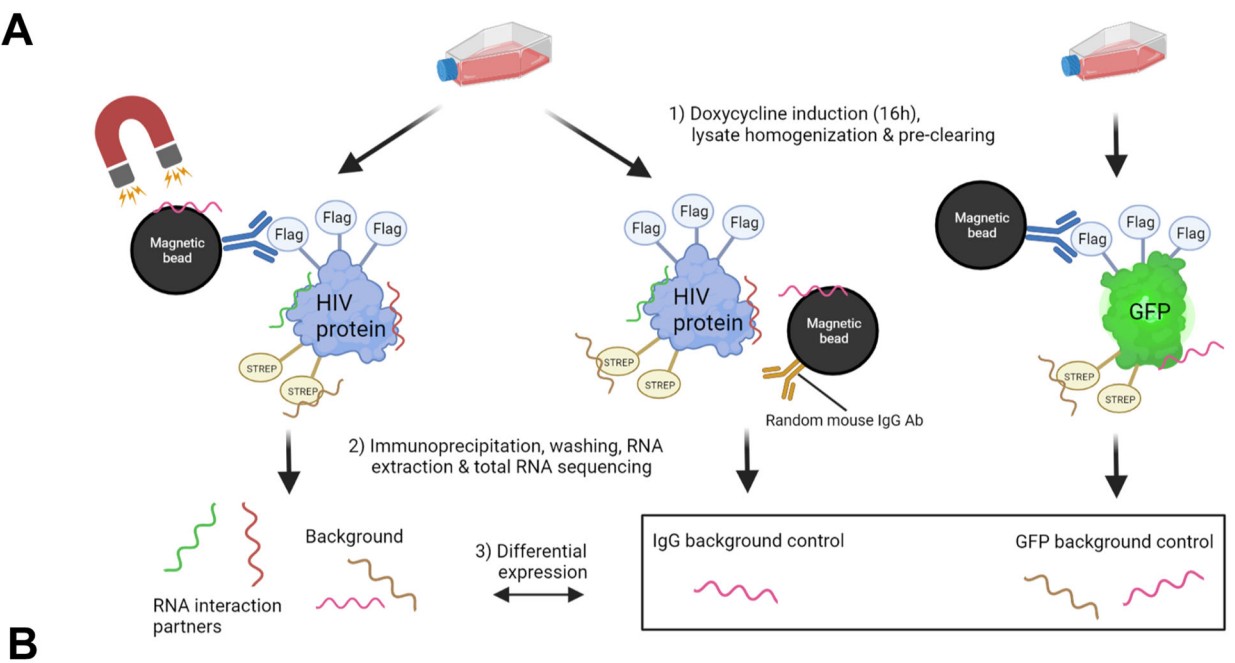

AFF2 and AFF3 in the Tat nRIPseq samples compared to the IgG and GFP control samples (Appendix Fig. S8).

Gene ontology terms enriched for HIV-1 proteins Gag, Nucleocapsid, Rev, and Integrase are all related to the translation machinery of the host cell, including "cytoplasmic translation" (GO:0002181) and "structural constituent of the ribosome" (GO:0003735). For Matrix, the top enriched GO terms include "structural constituent of chromatin" (GO:0030527) and

◀ **Figure 1.  Native RNA immunoprecipitation and sequencing (nRIPseq) of HIV-1 proteins.**

(A, B) Overview of four nRIPseq assays (A–D) performed on cell lines that express a single 2xStrepTagII-TEV-3xFLAG-tagged HIV-1 protein. Assay A was performed on uninfected Jurkat T-Rex cells with anti-FLAG antibody. Assay B used uninfected Jurkat T-Rex cells and MagStrep "type 3" XT beads. Assay C and D were performed on uninfected and NL4.3 HIV-1 infected SupT1 cells, respectively, with anti-FLAG antibody. Background controls included nRIPseq with an anti-mouse IgG antibody (A, C, D) or unconjugated beads (B) and nRIPseq of 2xStrepTagII-TEV-3xFLAG-tagged GFP. (C) Number of detected RNA interactors for all HIV-1 proteins for each of the four performed assays. Gray depicts assays that were not executed for the corresponding protein. (D) Ratios of the biotypes of the detected RNA interactors for all HIV-1 proteins for each of the performed nRIPseq assays (A–D). NC nucleocapsid, MA matrix, IN integrase, PR protease, CA capsid.

"nucleosome assembly" (GO:0006334), as 43 out of 93 detected RNA interaction partners of Matrix were histone gene mRNAs. The few RNA interactions detected for Protease (9) did show significant enrichment for genes involved in the "positive regulation of canonical Wnt signaling pathway" (GO:0090263): WNK1, WNK2, and PPP2R3A. The canonical Wnt signaling pathway has been reported to inhibit HIV transcription and correlates with HIV controller status (Wallace et al, 2020). Finally, we attempted to perform GO analysis on interactors specifically enriched in the nRIPseq workflow for infected SupT1 cells (assay D). Although limited enrichment was found, for MA, there was a significant enrichment of GO terms actin cytoskeleton (GO:0015629), actin binding (GO:0003779), and myosin complexes (GO:0016459) (Appendix Table S7), indicative of reordering cellular structures to promote trafficking of viral proteins and complexes.

In order to unravel the mechanism behind the specific RNA binding of the HIV proteins, a motif discovery search was performed using the Hypergeometric Optimization of Motif EnRichment (HOMER2) software. This rendered eight enriched de novo RNA motifs in the RNA sequences of the Matrix interactors, from which the top three are shown in Fig. 4B. For the other proteins, no enriched motifs could be identified in the RNA sequences of the detected interactors.

## ASO-mediated knockdown screening for prioritization of functional HIV-1 protein binding RNA interactors

In the past, several large loss-of-function screens were performed to identify host dependency factors (HDFs) or restriction factors (RFs) in HIV-1 replication by König et al, (2008); Brass et al, (2008); Yeung et al, (2009); Zhou et al, (2008) and Hiatt et al, (2022). We overlapped the top 373 identified hits (interaction score >50) with these loss-of-function screens and the prior established HIV-1 protein–host protein interactome of Jäger et al, (2011) and found a limited but significant overlap with Zhou et al, (2008) ($p = 0.015$) and Brass et al, (2008) ($p = 0.027$), comparable to the poor mutual overlap between the RNAi screens themselves (Fig. 4C). The overlap between the nRIPseq RNA interactors and all these screening efforts are available in Dataset EV2.

To assess to which extent the identified RNA interaction partners might be functional as an HDF or RF, we performed an in vitro antisense oligonucleotide (ASO; gapmer) mediated knockdown in SupT1 cells on a selection of 15 RNA interactors across 7 HIV-1 proteins and assessed the effect on GFP-tagged NL4.3 HIV-1 infectivity (Appendix Fig. S9). For three RNA interactors, we observed significantly impaired HIV-1 infectivity when the SupT1s were pretreated for 48 h with two target-specific ASOs, compared to a condition treated with a scrambled control ASO (Fig. 5A). These three are ALF transcription elongation factor 2 (AFF2) mRNA interactor of

Tat, 60S acidic ribosomal protein P0 (RPLP0) mRNA interactor of both Nucleocapsid and Rev, and H4 clustered histone 9 (H4C9) mRNA detected as interacting with Nucleocapsid, Gag, and Rev. Knockdown was confirmed with qPCR both before and 48 h post HIV-1 infection (Fig. 5B). In addition, cell viability upon knockdown and HIV-1 infection was assessed via propidium iodide staining and flow cytometry at 24 and 48 h post infection (Appendix Fig. S10), This indicated that the cell viability for the ASO-treated conditions was within the range of the ASO-scrambled control, with an exception of RPLP0 ASO2.

Furthermore, we explored the potential effect of these three mRNAs on HIV-1 latency reactivation. For this we performed ASO-mediated knockdown in the J-Lat 10.6 HIV-1 reactivation cell line model with and without the addition of an HDAC inhibitor (SAHA) and performed qPCR readout for knockdown and measured GFP signal via flow cytometry as a proxy for HIV-1 reactivation (Appendix Figs. S11, S12). Overall, there was only a limited knockdown observed for AFF1 ASO1 and good knockdown for H4C9 ASO1, 20 and 80% knockdown, respectively. However, despite the fact the positive control with PMA reactivated J-Lat 10.6 to around 40%, we could not detect an effect on reactivation in the ASO-treated conditions that was different from the scrambled ASO control.

## Discussion

Here, we present four nRIPseq workflows across two cell types and infection states, to explore HIV-1 protein–host RNA interactions across all HIV-1 (poly)proteins, identifying 1727 protein–RNA interactions in the Jurkat cell line and an additional 1558 interactions in the Supt1 cell line for a selected set of proteins. In terms of overlap between RNA interactors, ~24% of identified RNA interactors were found in at least two out of four nRIPseq workflows for their respective HIV-1 protein (NC, MA, Tat, Rev, Gag, Pol), generating a valuable list of potential targets across cell lines and HIV-1 infection status (Fig. 2A). Of note, we did observe differences in the number RNA interactors being identified uniquely with one nRIPseq workflow, this might be partially explained by the different masking of the RNA binding place of the protein and sterical hindrance by the alternative FLAG or STREP-based nRIPseq method together with a more promiscuous binding of the FLAG antibody. In addition, the inherent differences between Jurkat and SupT1 cell lines in terms of origin (T cell leukemia vs. T cell lymphoblastic lymphoma), cellular steady-state transcriptomes, and phenotype of the cells (e.g., membrane markers), also influence the wiring of cellular processes such as antiviral defenses (e.g., differences in interferon response), making that these cell models can behave differently to the expression of HIV-1 proteins or to HIV-1 infection.

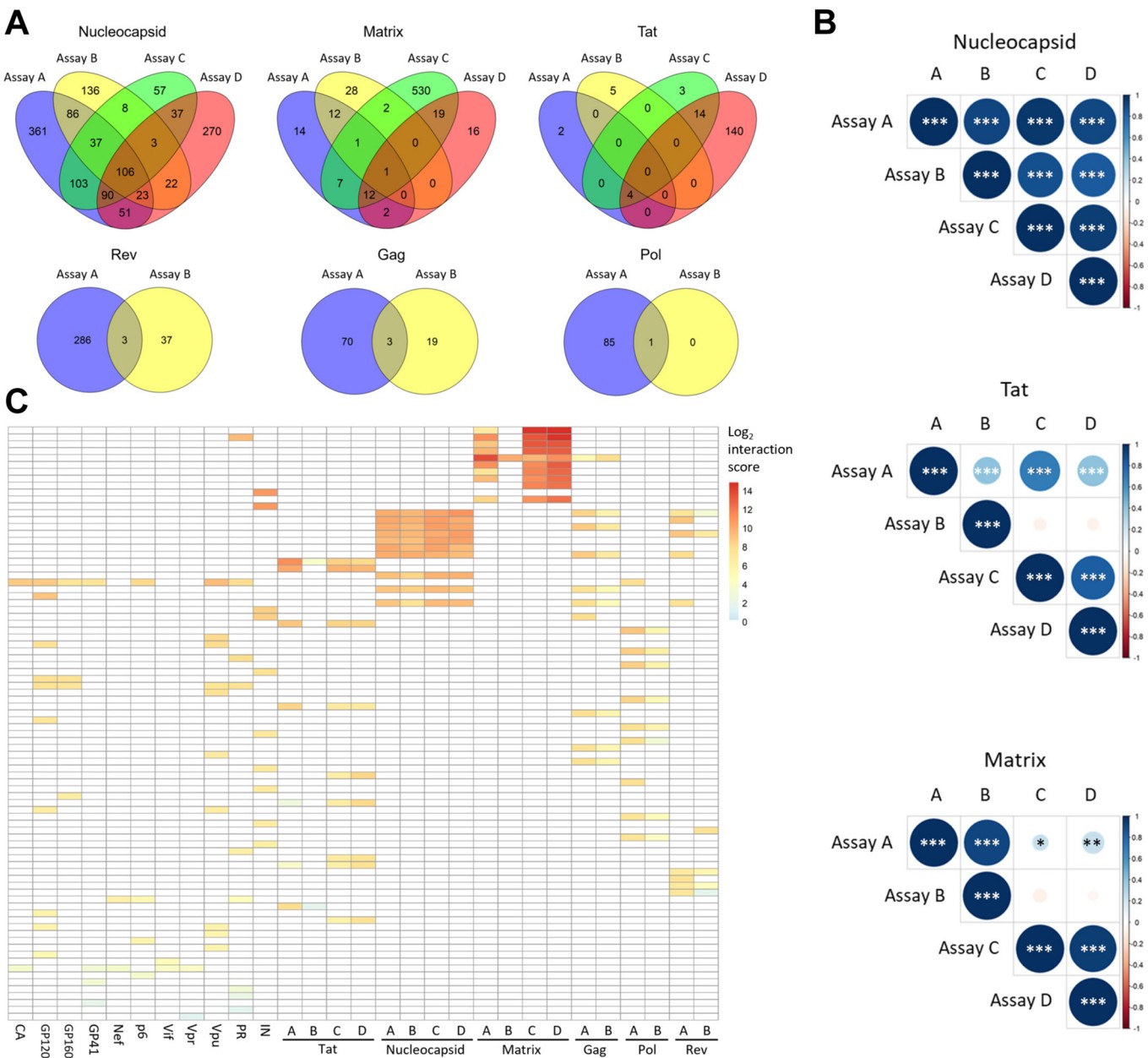

**Figure 2. Overlap and correlation between HIV-1 protein–RNA interactors identified by four nRIPseq workflows.**

(A) Venn diagrams showing the overlap between the RNA interactors identified by the four nRIPseq assays (A–D) for the HIV-1 proteins Nucleocapsid, Matrix, and Tat. For Rev, Gag, and Pol, an overlap between assays A and B is shown. (B) Spearman correlations between the interaction scores of the four nRIPseq assays for Nucleocapsid, Tat, and Matrix. Positive and negative correlations are depicted in blue and red, respectively. Significant correlations are indicated with asterisks and $p$ values are derived from the Spearman correlation analysis, all correlation results and $p$ values can be found in Appendix Table S5 ($p < 0.05$:*, $p < 0.001$: **, and $p < 0.0001$: ***). NC assay comparison $p$ values were all 0. Tat assay comparison $p$ values: A–B 0.000179, A–C 2.79E-21, A–D 9.08 E-07, B–C 4.25E-01, B–D 4.57E-01, C–D 5.2 E-35. Matrix assay comparison $p$ values: A–B 3.78E-208, A–C 4.66E-03, A–D 5.69E-07, B–C 6.88E-02, B–D 2.94E-01, C–D 1.61E-264. (C) Heatmap showing the top ten detected interactors of each HIV-1 protein, based on the mean interaction score. Negative scores are depicted in white.

## (Non-coding) RNAs interacting with the HIV-1 proteome

In general, the clear majority of the identified RNA interactors of HIV-1 proteins are protein-coding mRNAs (86.1%), although nRIPseq using preconjugated MagStrep "type3" XT beads instead of anti-FLAG antibody appears to provide a broader overview of the non-coding binding partners. A potential higher background

binding might mask these interactions in the anti-FLAG nRIPseq results, as non-coding RNAs are often very low expressed, needing deeper sequencing. Overall, we did, however, identify a number of strong HIV-1 protein–non-coding RNA interactions. We confirm multiple reports that Gag binds the non-coding 7SL RNA abundantly through its Nucleocapsid domain to retain this RNA in its virions (Kutluay et al, 2014; Simonova et al, 2019; Pham et al,

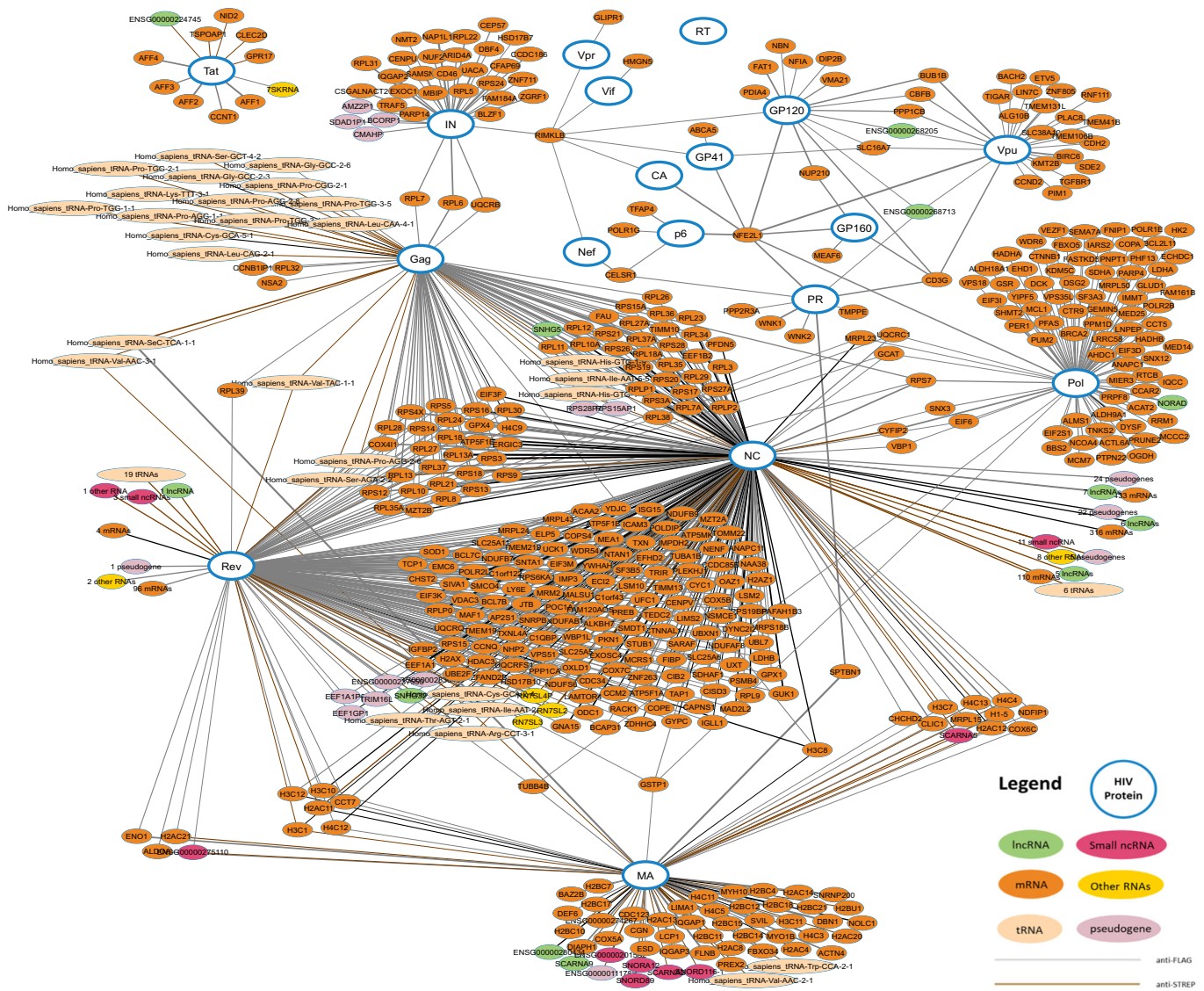

**Figure 3. Interaction network of HIV-1 proteins and host RNA interactors in the Jurkat cell line.**

In total 1754 interactions were detected between host RNAs and 17 HIV proteins in Jurkat cells by either anti-FLAG nRIPseq (gray), anti-Strep nRIPseq (brown) or both (black). Thicker connectors correspond to interactions with higher interaction scores.

2018; Bouwman et al, 2014). 7SL RNA is part of the signal recognition particle (SRP) ribonucleoprotein complex that direct mRNA translation through transmembrane pores. The reason for co-encapsulation of 7SL RNA by HIV-1 and other retroviruses is not well understood, but Wang et al showed that the HIV-1 restriction factor APOBEC3G is packaged into the virion through 7SL RNA interaction, showing that 7SL RNA aids in innate antiviral immunity (Wheeler et al, 2018; Simonova et al, 2019). The strongest lncRNA–HIV-1 protein interactions detected were: (1) NORAD (non-coding RNA activated by DNA damage) interacting with Pol, (2) an interaction between Matrix and a novel PARVB sense overlapping lncRNA (ENSG00000280434), and (3) lncRNA GIHCG interacting with Nucleocapsid. None of these three lncRNAs have been described or explored in the context of an HIV-1 infection and this database could thus be mined as a source

for functional characterization studies of new HIV-1 related lncRNAs. We also identified multiple pseudogene interactors. As the classification of pseudogenes as non-functional has recently been challenged (Luo et al, 2012; Core et al, 2008; Skalska et al, 2017; Statello et al, 2021), one should not completely ignore these interactions, as they might as well have a functionality, although the interaction could also arise from a shared RNA sequence/motif with the pseudogene's functional parental gene.

## The nucleocapsid domain contributes to Gag-host RNA binding

We were the first to explore HIV-1 protein–host RNA interactions across all HIV-1 (poly)proteins, but a few studies had however already performed analogous experiments for single HIV-1

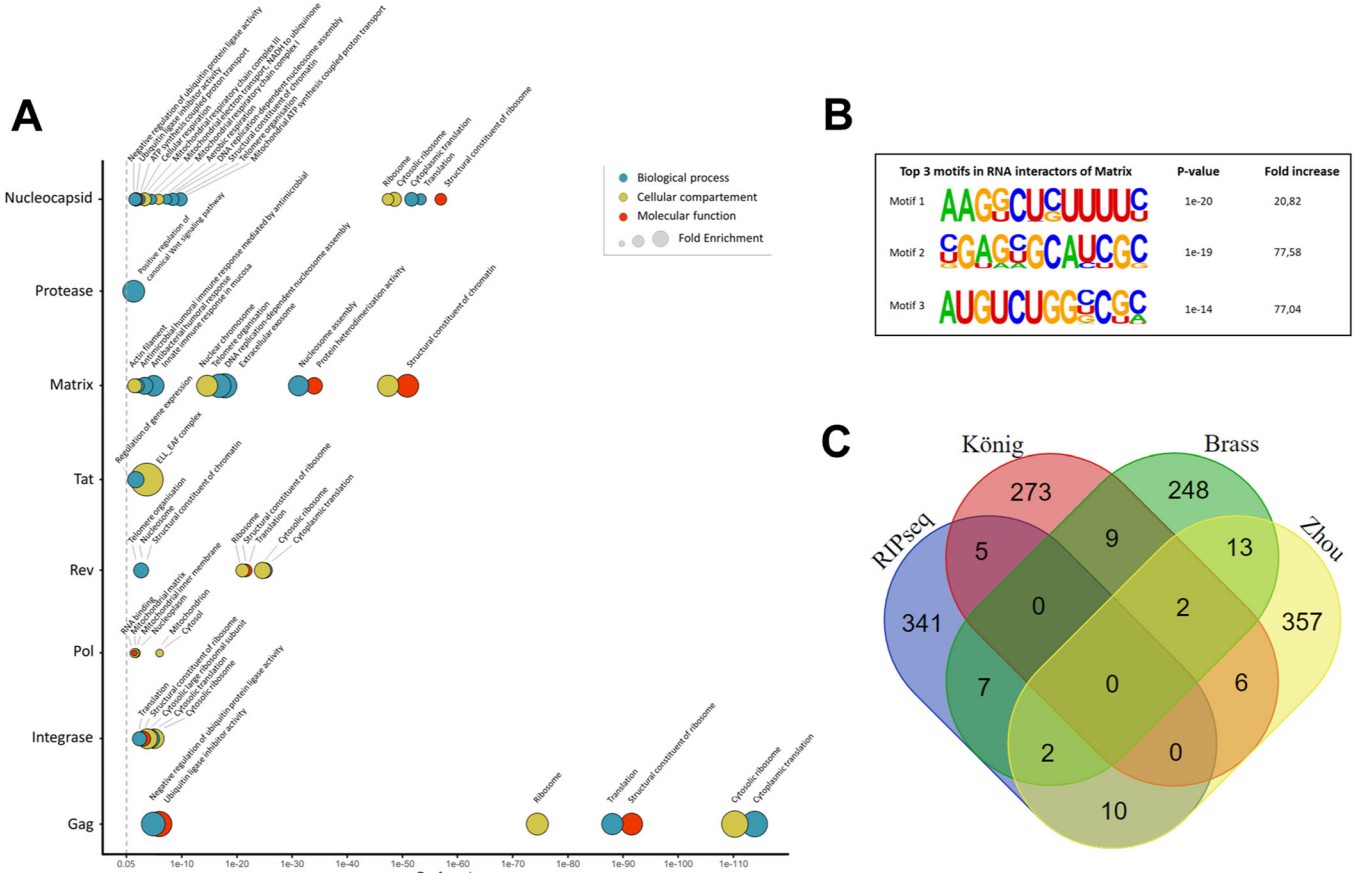

**Figure 4. Analysis of the HIV-1 protein-host RNA interactors.**

(A) Gene ontology (GO) enrichment analysis was performed with DAVID (Sherman et al, 2022) for the RNA interaction partners of every protein detected in the Jurkat cell line. The size of the circle represents the fold enrichment of the GO term. For the sake of clarity, only the top five of each DAVID functional annotation cluster is depicted (full table in Appendix Table S6). P values are Bonferroni corrected and generated by the DAVID software analysis and resulting from Fisher's Exact testing. A gray dashed line shows the 0.05 significance cutoff. (B) The top three enriched motifs identified with HOMER2 in RNA interaction partners of Matrix detected in the Jurkat cell line. Fold increase is calculated compared to the background of non-interacting RNAs (with average sequencing counts >500). (C) Venn diagram showing the overlap between detected interaction partners with enrichment scores >50 and host dependency factors identified by the large-scale siRNA screens performed by König et al, (2008); Brass et al, (2008) and Zhou et al, (2008).

proteins for which we could confirm a number of results. Kutluay et al, (2014) performed crosslinking immunoprecipitation (CLIP) sequencing to assess RNA binding by Gag. They showed that, while the Nucleocapsid domain of Gag binds with the cis-acting packaging element Psi for selective packaging of the HIV-1 RNA genome, there is also host RNA-Gag binding that serves as a scaffold for Gag monomers during initial multimerization for virion formation. This host RNA binding is semi-specific, as they report a preference for AG-rich RNA binding before virion assembly, and a shift towards GU-rich RNAs after assembly and maturation. Because we relied on nRIPseq rather than the more demanding CLIP assay for the sake of throughput, we lacked the information on nucleotide enrichment in our dataset. This also explains our difficulty in identifying specific RNA motifs (only for the Matrix domain we could identify eight de novo motifs): nRIPseq sequences the complete RNA interactor rather than the binding site, as achieved by CLIPseq. But we can confirm that Gag binds numerous host RNAs, potentially driven by its Nucleocapsid

domain, as the majority of identified Gag interactors are shared with the NC subunit (Fig. 3). Of note, the individual Gag subuntis NC and MA are cleaved and generated only during or after budding of the HIV-1 particle. Hence, the RNA interactors found for the cleaved MA and NC proteins should be interpreted in this context.

### Specific tRNA-Gag interactions: tRNA^Lys3 (UUU), tRNA^Lys1,2 (CUU), tRNA^SeC1 (TCA)

Kutluay et al, (2014) also reported that Gag binds a multitude of host tRNAs with its M Source data are available online for this figure.atrix domain, shown to redirect Gag towards the plasma membrane, rather than intracellular membranes, to facilitate budding. Our data confirms that Gag has a high preference for tRNA binding, including: (1) tRNA^Lys3 (UUU), the primer used by HIV-1 during reverse transcription (Zhang, 2021; Telesnitsky and Wolin 2016), (2) its non-primer lysine tRNA isoacceptor tRNA^Lys1,2 (CUU) was also found to bind Nucleocapsid, a

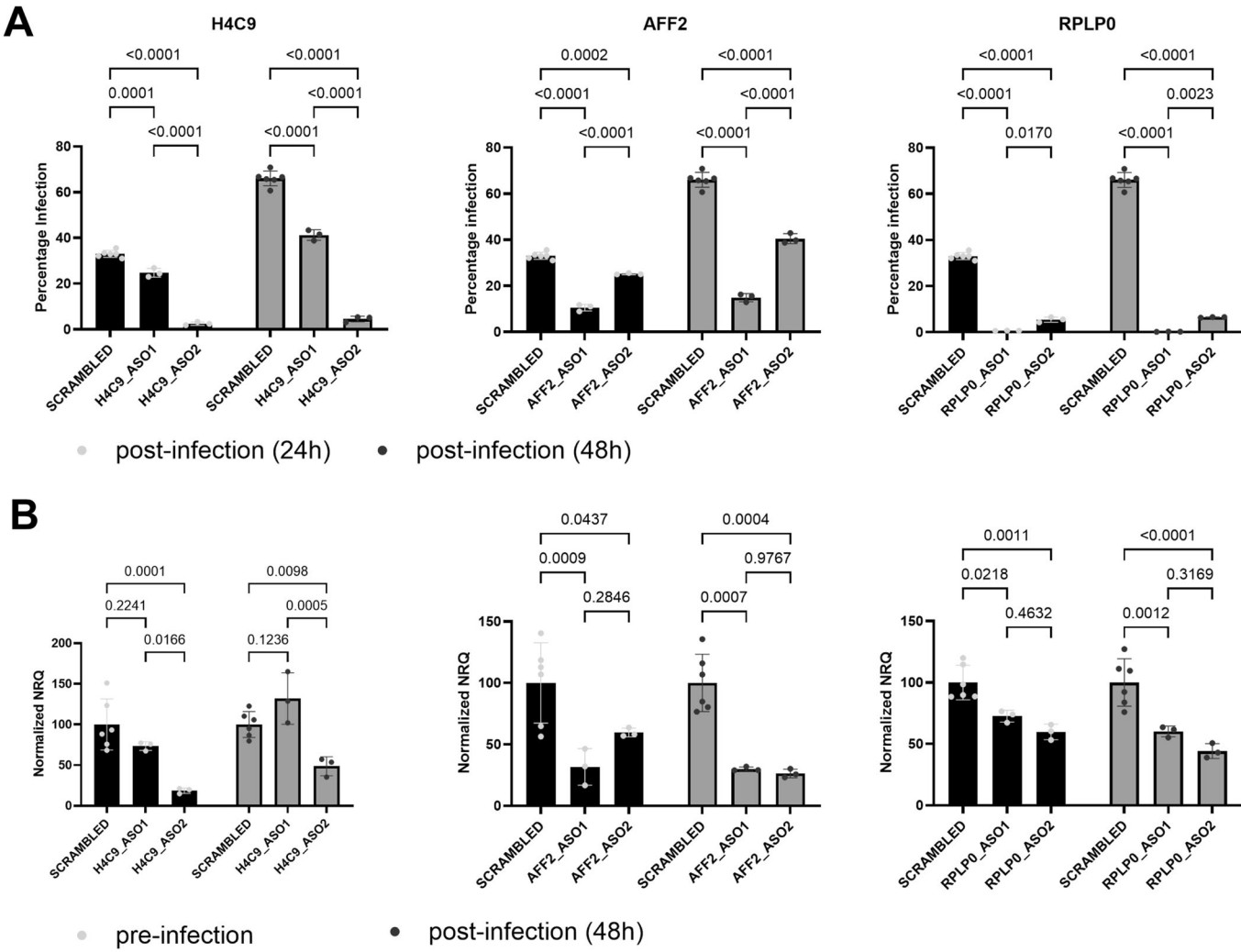

**Figure 5. Effect of antisense oligo (ASO) mediated knockdown on HIV-1 infectivity.**

SupT1 cells were treated for 48 h with ASOs that specifically target H4C9, RPLP0, or AFF2 mRNA for degradation, before the cells were infected with GFP-tagged NL4.3 HIV-1. (A) Percentage of GFP expressing cells, corresponding to the number of HIV-1 infected cells, 24 and 48 h post infection in cells treated with specific ASOs or with a scrambled non-targeting control ASO. (B) Normalized quantities of RNA expression relative to the condition treated with the scrambled control ASO, 48 h post ASO treatment and 48 h post infection. The experiment was performed once with three technical replicates for the on-target ASO-treated conditions and six technical replicates for the scrambled controls. Statistical testing was performed using a two-way ANOVA test in GraphPad Prism 10.2.3. Source data are available online for this figure.

tRNA that had been reported to be preferentially encapsulated in the HIV-1 virion as well, although it is not used as a primer (Telesnitsky and Wolin 2016; Pavon-Eternod et al, 2010; Kleiman et al, 2010), and (3) 34 non-lysyl tRNAs. In contrast to Kutluay et al, (2014), we observed that Matrix binds tRNAs to a lesser extent when it is not included as a domain in the Gag polyprotein, suggesting that Gag-tRNA binding might not be dominated by the Matrix domain alone. Indeed, Kleiman et al showed that tRNA$^{Lys}$-Gag binding occurs rather through interactions between the Capsid C-terminal domain, Lysyl-tRNA synthetase, and tRNAs (Kleiman et al, 2010). One of the few tRNA interactors of Matrix we did find was tRNA$^{SeC1\ (TCA)}$, with an abundancy of up to 1000 times higher than other tRNA interactors in this dataset. tRNA$^{SeC1\ (TCA)}$ is the carrier molecule for the 21st amino acid Selenocysteine (Sec; only one selenium atom different from cysteine) and has some unique features: it recognizes the UGA codon, which normally serves as a stop codon, but in particular,

circumstances, triggered by Sec availability and the presence of a selenocysteine insertion sequence (SECIS), is used by tRNA$^{SeC1\ (TCA)}$ to insert a Sec amino acid (Berry et al, 1993). Interestingly, selenium deficiency has been negatively correlated with survival for those living with an HIV-1 infection (Baum et al, 1997), and the levels of host selenoproteins decrease during an HIV infection (Gladyshev et al, 1999; Guillin et al, 2022). The sequestration of tRNA$^{SeC1\ (TCA)}$ by Matrix could explain this decrease. tRNA$^{SeC1\ (TCA)}$ has a unique, divergent primary and secondary structure which might be used for specific Matrix binding. Besides Gag and Matrix, we observed also extensive tRNA binding for Rev, which was not reported before. Further research will have to examine the biological function of these numerous observed Rev-tRNA interactions, but this additional tRNA-Rev binding could explain why Simonova et al observed 770 tRNA copies per HIV-1 virion, far higher than expected (Simonova et al, 2019).

## Tat binds members of SEC as mRNAs

Tat is a known RNA binding protein, as it recruits P-TEFb to stalled viral transcripts by association with both 7SK RNA and the TAR motifs of the viral transcripts (Pham et al, 2018). Only one study, by Bouwman et al, (2014), also examined its interaction with other cellular RNAs by performing Tat CLIP coupled with microarray analysis, reporting 317 Tat-mRNA interactions (Bouwman et al, 2014). We only detected three of these interactions in this study. This limited overlap might be explained by the other nature of the used assays, as native and crosslinking conditions will reveal different interactions based on their stringency (Wheeler et al, 2018). One remarkable observation we made was that Tat binds, besides 7SK RNA, a number of mRNAs of proteins involved in the super elongation complex (AFF1, AFF2, AFF3, AFF4, and Cyclin T). This was discovered in two different cell lines and multiple independent experiments. A large number of studies have shown that Pol II transcription pausing and unpausing by the P-TEFb containing super elongation complex (SEC) is a major eukaryotic transcription regulation mechanism, involved in many human diseases and conditions (Fujinaga, 2020; Luo et al, 2012; Core et al, 2008; Muse et al, 2007). HIV-1 transcription only really gets going when Tat levels are sufficiently high to recruit the SEC to viral TAR-regions. The observed binding of Tat with the mRNAs of protein subunits of the SEC, indicates that their nascent transcription might be regulated, similar to Tat transcription, in a feed-forward loop, increasing the levels of the subunits of the SEC, to meet the increasing demand during HIV-1 infection (Skalska et al, 2017).

## Inhibitory effect on HIV-1 replication after knockdown of HIV-1 protein-interacting RNAs: AFF2, H4C9, and RPLP0

This dataset of HIV-1 protein-interacting RNAs shows a limited overlap with HDFs or RFs identified by five large loss-of-function screens and the prior established HIV-1 protein–host protein interactome (Brass et al, 2008; Zhou et al, 2008; König et al, 2008; Yeung et al, 2009; Hiatt et al, 2022; Jäger et al, 2011). But one has to consider that the individual overlap between these loss-of-function screens was small as well, probably fueled by the high false positive and false negative rate associated with these types of screens or differences in the cell-based systems used (Zhu et al, 2014). Additionally, recent reports have shown that nascent RNA often interacts with regulatory proteins to regulate gene transcription, to the extent that all pre-mRNAs may be considered as 'bifunctional RNAs', possessing both coding an regulatory functions (Skalska et al, 2017). Therefore, we continued to explore our hypothesis that these HIV-1 protein-interacting mRNAs might be acting as HIV-1 HDFs or host restriction factors, which are either transcriptionally regulated by the interaction or regulating through the interaction. Antisense oligo-mediated knockdown of three mRNAs (AFF2, H4C9, and RPLP0) resulted indeed in a significantly inhibited HIV-1 infectivity in vitro. However, RNA knockdown cannot distinguish between an effect due to RNA or protein knockdown, as these are strongly related. Therefore, follow-up experiments should investigate whether the loss of RNA expression (or the possible induced loss in protein expression) is responsible for the observed phenotype. It does provide, however, further evidence that AFF2 plays an essential role during HIV-1 infection. Luo et al reported that AFF2 is part of SEC-like2, comparable to the SEC (containing

AFF1 or AFF4), but without confirmed presence of ELL- or EAF-related proteins. Our results suggest that AFF2 presence is needed for an efficient HIV-1 infection, either via the actions of SEC-like2 or an alternative route. Altogether, our small-scale ASO-mediated screen to probe the functional effects of HIV-1 protein-interacting RNAs on HIV-1 replication indicates that this HIV-1 protein–host RNA dataset is a valuable source for HIV-1 research.

## Limitations

This study has some limitations. First of all, the interactome was determined in cell lines rather than primary CD4 + T-cells, to meet the large number of cells needed for nRIPseq. The FLAG- and Strep-tag attached to the HIV-1 in these cell lines allowed, however, for more efficient pull down, essential for successful nRIPseq. Moreover, the cell lines are the same as the ones used in a previously published large-scale HIV-1 protein–host protein interaction study (Jäger et al, 2011) and represent primary cells better than the HEK293T or HeLa cell lines used in the large-scale loss-of-function screens for HIV-1 host factors (Brass et al, 2008; Zhou et al, 2008; König et al, 2008; Yeung et al, 2009; Hiatt et al, 2022). Second, nRIPseq workflows are challenging protocols that can be prone to a higher false positive rate when the proper background controls are not in place, causing variability between experiments and setups. In addition, to investigate RNA interaction partners in the context of HIV-1 infection, the SupT1 system was engineered with a constitutive expression of the HIV-1 protein of interest, rather than the doxycycline inducible Jurkat system. This difference could also contribute to the observed differences in identified interaction partners between assay A/B and C/D. However, this has been countered by including multiple background controls (GFP and IgG) and by the usage of STREP and FLAG-based nRIPseq workflows that allowed to identify high-accuracy interactions in their overlap. Third, to validate the identified RNA interaction partners and support that loss of RNA expression (and the RNA molecule itself) is responsible for the observed phenotype, follow-up experiments are mandatory that integrate protein level expression. Fourth, the presented cellular systems use tagged HIV-1 proteins for nRIPseq procedures that are overexpressed at (likely) non-physiological levels. This could lead to miss-aggregating or miss-trafficking of these proteins within the cell, obscuring the identification of true RNA interactors. Follow-up pulldown experiments should ideally attempt to use HIV-1 protein-specific antibodies during an active HIV-1 infection.

# Conclusions

To conclude, we constructed a new extensive HIV-1 protein–host RNA interactome across all HIV-1 proteins with several variants of the nRIPseq platform in two cell lines and HIV-1 infection states. The considerable overlap suggests a high accuracy of the identified interactions. This study gives a sneak peek into the pathways and HIV-1 replication steps regulated through these interactions, like the potential regulation of the super elongation complex on a post-transcriptional level by Tat. Finally, this interactome will be available to use as a resource for further research and prioritization of cellular RNA interactors affecting HIV-1 infection or replication.

# Methods

## Cell line generation, cell culture, and HIV-1 infection prior to RIPseq

Jurkat Trex cell lines expressing a single 2xStrepTagII-TEV-3xFLAG-tagged HIV-1 protein were created by Jäger et al, (2011), as previously described. Briefly, they cloned codon-optimized versions of HIV-1 open reading frames (ORFs) into the pcDNA4/TO vector (Invitrogen) carrying a 3′ 2xStrepTagII-TEV-3xFLAG sequence. The ORFs of PR and Pol were catalytically inactivated by D25N exchange and an influenza HA signal peptide was added 5′ to the GP41 coding sequence. Transfection of Jurkat Trex cells (Invitrogen) with the linearized vector, Zeocin selection, and limited dilution generated stable monoclonal cell lines. Prior to immunoprecipitation, construct expression was induced with 1 µg/mL doxycycline stimulation for 16 h. SupT1 cell lines were newly generated by transfecting SupT1 cells (NIH HIV-1 reagent program) with the linearized (Sca-I) vector using JetOptimus (Polyplus). Limited dilution and Zeocin selection produced stable monoclonal cell lines that (without doxycycline induction) constitutively express a single 2xStrepTagII-TEV-3xFLAG-tagged MA, NC, or Tat protein. Jurkat Trex cell lines were cultured in RPMI (Gibco), supplemented with 10% FCS (HyClone), Pen/Strep, sodium-pyruvate (1 mM; Gibco), HEPES (15 mM; Gibco), Blasticidine (10 µg/mL; InvivoGen), Zeocin (300 µg/mL; InvivoGen), at 5% $CO_2$ and 37 °C. SupT1 cell lines were cultured in RPMI, supplemented with 10% FCS, Pen/Strep, sodium-pyruvate (1 mM), HEPES (15 mM), Zeocin (150 µg/mL), at 5% $CO_2$ and 37 °C. HIV-1 infection of SupT1 cells prior to RIPseq was done with VSV-G pseudotyped NL4.3 HSA-IRES-Nef dEnv labstrain virus by spinoculation for 90 min (2300 rpm, 32 °C). Cells were harvested 24 h post infection with ~65% infection rate confirmed by flow cytometry.

## RNA immunoprecipitation sequencing worfklows (RIPseq)

For RNA immunoprecipitation sequencing (RIPseq) 20 million snap-frozen cells per replicate were lysed in 3 mL lysis buffer (20 mM Tris pH 8.0, 200 mM NaCl, 2.5 mM $MgCl_2$, 0.00025% Triton X-100 in DEPC-treated water, freshly supplemented with 60 U/mL SUPERase-In (Thermo Fisher), cOmplete Protein Inhibitor Cocktail (Roche) and 1 mM Dithiothreitol). Lysates were vortexed vigorously, incubated 30 min on ice with regular agitation, and centrifugated for 10 min (15,000 × g, 4 °C). The supernatant was precleared with 30 µL protein G Dyna beads (Thermo Fisher) for 1 h at 4 °C on a tube revolver rotator. The precleared lysate was incubated overnight (on a tube revolver rotator at 4 °C) with 10 µg mouse/IgG2b DYKDDDDK Tag (FLAG) Monoclonal FG4R Antibody (Invitrogen, MA1-91878), followed by a 4 h incubation in the presence of a 100 µL protein G Dyna beads. The beads were collected magnetically, washed six times, and resuspended in 55 µL lysis buffer. About 5 µL of the lysate was analyzed by 4–12% SDS-PAGE (Invitrogen). The rest of the lysate was supplemented with 5 µL Proteinase K (Thermo Fisher) and 95 µL Proteinase K buffer (10 mM Tris-HCl, 100 mM NaCl, 1 mM EDTA, 0.5% SDS, and 60 U/mL SUPERase-IN in DEPC-treated water) and incubated for

45 min at 50 °C and 10 min at 95 °C. The RNA fraction was extracted with the miRNeasy kit (Qiagen) using the manufacturer's instructions, including on-column DNase digestion and elution in 14 µL nuclease-free water. Total RNA library preparation was performed using the SMARTer Stranded Total RNA-Seq Mammalian Pico v2 Kit (Takara) using the manufacturer's instructions with 8 µL RNA input, 4 min fragmentation, and 16 PCR2 cycles. Quality control was done using a Fragment Analyzer System (Advanced Analytical) and qPCR KAPA quantification (Illumina) was performed to allow normalized pooling of the samples. The pool was paired-end sequenced using a NovaSeq S1 flow cell, generating approximately 50 million reads per sample.

RIPseq was performed in batches of 3 HIV-1 protein Ips and 1 FLAG-tagged GFP background control IP (all in triplicate). For each HIV-1 protein IP also, a background control IP was included in triplicate using 10 µg anti-mouse IgG rabbit IgG antibody (Thermo Fisher, no. 11895905). For Streptavidin-based RIPseq, a similar procedure was performed using 50 µL preconjugated MagStrep "type3" XT beads (iba, no. 2-4090-002) for IP and unconjugated MagStrep "type3" XT beads for preclearance and background control.

## Interactome construction and bioinformatic analysis

RNA sequencing quality control was performed using FASTQC in Python (version 3.6.6) and reads were mapped to the human reference genome (CRCh38 build) using STAR, followed by calculation of the counts per gene using HTSeq. Differential expression on normalized counts was assessed using the DESeq2 package (1.30.0) in R (64 Bit, version 4.0.3) by comparing HIV-1 protein nRIPseq samples simulaneously with the corresponding anti-mouse IgG and GFP background controls. Results were visualized with a volcano plot using the Enhanced Volcano package (1.8.0).

The foldchange cutoff $F_j$ and $P$adj-cutoff $P_i$, to accept a gene r (within the geneset $R_p$), with $P$adj-value $P_{p,r}$ and foldchange $F_{p,r}$, as RNA interaction partner of an HIV-1 protein p, were individually determined per protein p by optimizing the signal $n_H$ to background $n_B$ proportion, through maximizing $S_{p,i,j}$ with a background penalty score q of 5, as follows:

$$S_{p,i,j} = n_{H,i,j} - q \cdot n_{B,i,j}$$

$$\text{with } n_{H,i,j} = \left| \{ r \in R_p : (P_{p,r} < P_i \,\&\, F_{p,r} > F_j) \} \right|$$

$$\text{and } n_{B,i,j} = \left| \{ r \in R_p : (P_{p,r} < P_i \,\&\, F_{p,r} < -F_j) \} \right|$$

$$\text{with } P_i < 0.05 \,\&\, F_j \geq 2$$

$$\text{and } n_{B,i,j} < 0.05 \cdot n_{tot}$$

$$\text{with } n_{tot} = \left| \{ r \in R_p : (P_{p,r} < 0.05 \,\&\, F_{p,r} < -1) \} \right|$$

No more than 5% of the total background $n_{tot}$ was allowed. Signal-to-background optimization was visualized with the pheatmap package (1.0.12) in R (Appendix Fig. S5).

Interaction scores were calculated by substracting the average RPM of the two corresponding background controls from the RPM of the corresponding HIV-1 protein IP sample. The final interaction score of an RNA interaction partner was defined as the average binding score over all three replicates. Interaction networks were constructed and visualized using Cytoscape (3.8.0).

The identified RNA interaction partners per protein were used as input for the webtool DAVID 2021 (Sherman et al, 2022) for gene ontology enrichment analysis with GO BP_DIRECT, CC_DIRECT, and MF_DIRECT. The top five terms of each significant functional annotation cluster with a Bonferonni corrected $p$ value <0.05 were retained.

RNA motif enrichment analysis was performed with the Hypergeometric Optimization of Motif EnRichment (HOMER2) software (Heinz et al, 2010) with the full transcripts of the RNA interactors of a specific HIV-1 protein as input. The custom background setting was specified as all non-interacting RNAs with average nRIPseq sequencing counts >500.

### Antisense oligo knockdown screen and qPCR knockdown validation

SupT1 and J-Lat 10.6 cells (NIH HIV-1 reagent program) were cultured in RPMI medium (Gibco), supplemented with 10% FCS (HyClone) and Pen/Strep. SupT1 cells or J-Lat 10.6 were treated for 48 h with 5 μM of either a scrambled antisense oligo (ASO) (GTGCTGTTCCCGGGGA) or an ASO that targets a specific gene with three locked nucleic acids at either sides of the ASOs. J-Lat 10.6 cells were treated with AFF2, H4C9, and RPLP0 targeting ASOs. Three ASOs per target gene were designed with the LNCASO online webtool (sequences in Appendix Table S8). After 48 h, while keeping the ASO concentration at 5 μM, the SupT1 cells were infected with NL4.3 GFP-tagged HIV-1 labstrain virus by spinoculation for 90 min at 2300 rpm and 32 °C. Twenty-four hours and 48 h post infection, HIV-1 infection (SupT1) or HIV-1 reactivation (J-Lat 10.6) was assessed by flow cytometry readout of GFP whereas cell viability was assessed via proprium iodine staining. The normalized relative quantities of the target gene were determined with quantitative PCR 48 h post ASO treatment and 48 h post infection. RNA was extracted with the RNA innuPREP Mini Kit (Analytik, Germany) and reverse transcribed using the qScript cDNA Supermix (Quantabio, MA, USA), following the manufacturer's protocol. About 20 ng cDNA was used as input in a 10 μL quantitative real-time PCR (LightCycler 480 II, Roche Applied Science, Belgium). The SYBR Green kit (LightCycler 480 SYBR Green I Master, Roche Applied Science, Belgium) was used according to the manufacturer's protocol with 5 μL SYBR Green Master Mix and 250 nM primer concentrations (list of primers in Appendix Table S4). Each reaction was performed in duplicate. Cycling conditions of the LightCycler 480 (Roche Applied Science, Germany) were 95 °C for 5 min, 45 amplification cycles of 95 °C for 10 s, 58 °C for 30 s, and 72 °C for 30 s. Assay specificity was assessed visually in the LightCycler 480 software, based on the 60 °C to 95 °C melting curve and all samples past quality control. The stability of three (GAPDH, ACTB, and PLOD1) reference genes was verified with the geNorm method of Vandesompele et al, (2002) and the

normalization factors were determined by the geometric mean of their relative quantities. The primer efficiency was assessed on a two-fold cDNA standard curve (50–1.56 ng) and primers with efficiencies between 85–105% were retained. The normalized relative quantities (NRQs) of each gene were calculated using the (primer efficiency) ΔCt method.

## Data availability

The nRIPseq datasets produced in this study are made available at NCBI's Gene Expression Omnibus and are accessible through GEO Series accession number GSE268788 (https://www.ncbi.nlm.nih.gov/geo/query/acc.cgi?acc=GSE268788).

The source data of this paper are collected in the following database record: biostudies:S-SCDT-10_1038-S44319-024-00222-6.

## Peer review information

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

## Acknowledgements

TS received a strategic basic research fund from the Research Foundation—Flanders (grant number FWO 1S41219N); FWO-SBO SAPHIR grant (2020000501) funded WvS and provided a working budget; WT is funded by a doctoral assistant mandate (Department of Internal Medicine and Pediatrics, Ghent University); and LV is funded by the Research Foundation Flanders (grant number FWO 1.8.020.09.N.00). ViiV Healthcare supported this research with unrestricted grant PRI.DIV.2021.0017.01.

## Author contributions

**Tinus Schynkel**: Conceptualization; Resources; Data curation; Software; Formal analysis; Validation; Investigation; Visualization; Methodology; Writing—original draft. **Willem van Snippenberg**: Formal analysis; Investigation; Writing—review and editing. **Kimberly Verniers**: Formal analysis; Investigation; Writing—review and editing. **Gwendolyn M Jang**: Resources; Supervision; Writing—review and editing. **Nevan J Krogan**: Resources; Supervision; Writing—review and editing. **Pieter Mestdagh**: Conceptualization; Supervision; Funding acquisition; Writing—review and editing. **Linos Vandekerckhove**: Conceptualization; Supervision; Project administration; Writing—review and editing. **Wim Trypsteen**: Conceptualization; Supervision; Funding acquisition; Investigation; Methodology; Project administration; Writing—review and editing.

Source data underlying figure panels in this paper may have individual authorship assigned. Where available, figure panel/source data authorship is listed in the following database record: biostudies:S-SCDT-10_1038-S44319-024-00222-6.

## Disclosure and competing interests statement

The authors declare no competing interests. The funders had no role in the design of the study; in the collection, analyses, or interpretation of data; in the writing of the manuscript, or in the decision to publish the results.

