## [Peer Review File · EMBO Reports]

Interactome of the HIV-1 proteome and human host RNA

Tinus Schynkel, Willem Snippenberg, Kimberly Verniers, Gwendolyn Jang, Nevan Krogan, Pieter Mestdagh, Linos Vandekerckhove, and Wim Trypsteen

Corresponding author(s): Wim Trypsteen (Wim.Trypsteen@UGent.be) , Wim Trypsteen (Wim.Trypsteen@UGent.be), Linos Vandekerckhove (Linus.Vandekerckhove@UGent.be)

Review Timeline:

Submission Date:	9th Nov 23
Editorial Decision:	18th Dec 23
Revision Received:	17th May 24
Editorial Decision:	25th Jun 24
Revision Received:	18th Jul 24
Accepted:	23rd Jul 24

Editor: Achim Breiling

Transaction Report:

Dear Dr. Trypsteen,

Thank you for the submission of your research manuscript to EMBO reports. I have now received the reports from the three referees that were asked to evaluate your study, which can be found at the end of this email.

As you will see, the referees think that the findings are of interest. However, they have several comments, concerns, and suggestions, indicating that a major revision of the manuscript is necessary to allow publication of the study in EMBO reports. As the reports are below, and all the referee concerns need to be addressed, I will not detail them here, but, as is apparent from the reports, a better and more extensive validation of the identified mRNAs and interactions seems necessary to consider the manuscript for publication in EMBO reports.

Given the constructive referee comments, I would like to invite you to revise your manuscript with the understanding that all referee concerns must be addressed in the revised manuscript or in a detailed point-by-point response. Acceptance of your manuscript will depend on a positive outcome of a second round of review. It is EMBO reports policy to allow a single round of revision only and acceptance of the manuscript will therefore depend on the completeness of your responses included in the next, final version of the manuscript.

- 1) a .docx formatted version of the final manuscript text (including legends for main figures, EV figures and tables), but without the figures included. Figure legends should be compiled at the end of the manuscript text.
- 2) individual production quality figure files as .eps, .tif, .jpg (one file per figure), of main figures (up to 8) and EV figures. Please upload these as separate, individual files upon re-submission.

- 4) a complete author checklist, which you can download from our author guidelines

(<https://www.embopress.org/page/journal/14693178/authorguide>). Please insert page numbers in the checklist to indicate where the requested information can be found in the manuscript. The completed author checklist will also be part of the RPF.

5) that primary datasets produced in this study (e.g. RNA-seq, CHIP-seq, structural and array data) are deposited in an appropriate public database. If no primary datasets have been deposited, please also state this in a dedicated section (e.g. 'No primary datasets have been generated and deposited'), see below.

The accession numbers and database should be listed in a formal "Data Availability" section (placed after Materials & Methods) that follows the model below. This is now mandatory (like the COI statement). Please note that the Data Availability Section is restricted to new primary data that are part of this study. This section is mandatory. As indicated above, if no primary datasets have been deposited, please state this in this section

Data availability

8) Regarding data quantification and statistics, please make sure that the number "n" for how many independent experiments were performed, their nature (biological versus technical replicates), the bars and error bars (e.g. SEM, SD) and the test used to calculate p-values is indicated in the respective figure legends (also for potential EV figures and all those in the final Appendix). Please also check that all the p-values are explained in the legend, and that these fit to those shown in the figure. Please provide statistical testing where applicable. Please avoid the phrase 'independent experiment', but clearly state if these were biological or technical replicates. Please also indicate (e.g. with n.s.) if testing was performed, but the differences are not significant. In case n=2, please show the data as separate datapoints without error bars and statistics. See also: <http://www.embopress.org/page/journal/14693178/authorguide#statisticalanalysis>

9) Please also note our reference format:

10) We updated our journal's competing interests policy in January 2022 and request authors to consider both actual and perceived competing interests. Please review the policy <https://www.embopress.org/competing-interests> and update your competing interests if necessary. Please name this section 'Disclosure and Competing Interests Statement' and put it after the Acknowledgements section.

11) We now use CRediT to specify the contributions of each author in the journal submission system. CRediT replaces the author contribution section. Please use the free text box to provide more detailed descriptions and do not provide your final manuscript text file with an author contributions section. See also our guide to authors: <https://www.embopress.org/page/journal/14693178/authorguide#authorshipguidelines>

12) We would encourage you to use 'Structured Methods', our new Materials and Methods format. According to this format, the Materials and Methods section should include a Reagents and Tools Table (listing key reagents, experimental models, software and relevant equipment and including their sources and relevant identifiers) followed by a Methods and Protocols section in

which we encourage the authors to describe their methods using a step-by-step protocol format with bullet points, to facilitate the adoption of the methodologies across labs. More information on how to adhere to this format as well as downloadable templates (.doc or .xls) for the Reagents and Tools Table can be found in our author guidelines (section 'Structured Methods'):

13) Please remove the list of abbreviations from the manuscript and define each abbreviation upon first mention in the text.

14) Please order the manuscript sections like this, using these names:

Title page - Abstract - Keywords - Introduction - Results - Discussion - Materials and Methods - Data availability section - Acknowledgements - Disclosure and Competing Interests Statement - References - Figure legends - Expanded View Figure legends

I look forward to seeing a revised version of your manuscript when it is ready. Please let me know if you have questions or comments regarding the revision.

Yours sincerely,

Referee #1:

The manuscript by Schynkel et al. reports on the interactome of HIV proteins and host RNAs by using native RNA immunoprecipitation and sequencing (nRIPseq). The study provides a novel comprehensive resource for the HIV community. This is a carefully executed study with appropriate controls. To enrich for true interactors, the authors performed four assays, two with Jurkat T cells and in a subset two with SupT1 cells, uninfected and infected. However, the authors performed only very limited follow-up on only three targets, mRNAs and did not provide comprehensive validation on those mRNAs. It would strengthen the results of the manuscript by considering the following points:

- Jurkat T cells are treated with Doxycycline in comparison to the SupT1 cells. Dox treatment has been shown to influence the transcriptome profile. Do the authors suspect differences within Assay A and B to C (D)?
- Fig 1C and D: Why is the outcome of assay A to B so different
- Fig 2A: The assay result of D could be very informative, as here, the cells were infected. The authors should provide the identity of all overlapping interactors (in a table form for easy reference)
- Fig 2B: significance asterisks are hardly visible
- Fig 2C: please name the identity of the detected interactors (top 10 of each HIV protein), within this figure or in an extra table
- Fig 3 and 4a: please provide better resolution as text is blurry when zooming in
- Fig. 4c: please provide a table with the top 365 hits (identity) and respective overlap with screens
- Fig 5: the authors should delineate whether the RNA is responsible for the effect they see or whether the protein that is coded by the RNA is responsible? The authors should measure the protein level (does it change?), are there inhibitors, activators for the proteins known that could be used for these experiments? What actually leads to HIV inhibition upon ASO treatment?
- The authors consider the screening hits as RNAs potentially acting as host dependency "factors", however the authors should consider that these RNAs could possibly also aid in HIV defense. Based on the GO groups there are some that might lead into this direction, "innate immune response", "ubiquitin-ligase activity"
- The term host dependency "factor" for RNAs seems quite odd..
- The authors should compare their hits not only to the three mentioned siRNA screens, however also to the pooled shRNA screen (Yeung et al 2009) and CRISPR screens, e.g. Hiatt et al., Nat Comm 2022; Park et al., 2017; They could consider that the discovered RNAs might bind to a complex pulled-down by their approach, so possibly could bind in an indirect manner to host factors binding to HIV-proteins, and therefore might consider also proteomics studies such as Jager et al., 2011

Referee #2:

The manuscript by Schynkel et al. provides a useful resource for knowing the RNA interacting partners of the HIV-1 polyproteins. To identify the RNA interacting partners, the authors have performed native RIPseq in overexpressed viral proteins. The findings from this study can be useful for other studies and yet the reviewer is wondering if the authors can consider looking into the following points:

- Have the authors validated some of the interactions using qPCR?

- Could the authors explain a bit more about the differences in interaction in different cell lines? While the authors mention the masking of RNA binding site as a probable reason, which the reviewer strongly agrees with. But could there be other reasons? For both Assay A and C Flag-tag was used. While NP and Tat went down there was an enhancement in Matrix. Also, during virus infection of SupT1, NP and Tat show an increase, and probably some or all these RNA were identified in Assay A.
- Continuing on infection, could the authors look into what changes during infection between Assay C and D. More information on the types of RNA (coding or non-coding etc.) and the pathways affected? The biological processes have been focused more on non-infectious conditions.
- In the infection model have the authors investigated interaction with viral RNAs?
- Most of the identified RNA were protein-coding genes (86%). How the interactions of the 3 major HIV proteins with these coding RNA could influence downstream signaling processes. The authors have investigated 15 RNA interactors, with further interrogation on H4C9, RPLP0, and AFF2. There were 19 RNA that overlapped with either Brass et al. or Zhou et al, and it is not clear whether these were used for screening. The reviewer is curious to know the details of these overlapping RNAs, maybe as a supplementary table.
- In Fig5. For H4C9. Despite higher expression of ASO1, decreased infectivity. Have I missed something here?
- Can the authors attempt knockdown in an HIV reactivation model like Jlat or U1 to observe the effects on reactivation?
- Some Figures are difficult to read. In Fig2B. significant stars are difficult to read in the dark blue. Can consider changing them to white when it is a dark background. In Fig 4A font size should be increased considerably.

 Referee #3:

Here, Schynkel et al. carry out native RNA immunoprecipitation and sequencing (nRIPseq) to identify host cellular RNAs associated with HIV-1 proteins/polyproteins purified from cell lysates. They identify several hundred RNAs that co-precipitated with 18 individual tandem-affinity-tagged viral proteins. Nucleocapsid, Rev, Pol, and Gag pulled down the most RNA interactors. These RNAs were mostly mRNAs but tRNAs, lncRNAs, pseudogenes were also ID'd. To enhance rigor, samples were collected from two independent cell types in four total assay configurations, allowing the authors to identify a subset of RNAs that were reproducibly IP'ed and, thus, their top binders.

Interestingly, Tat pulled down mRNAs encoding components of a transcriptional super elongation complex that included CCNT1 and 7SK RNA, known regulators of the pTEF-b protein complex that is recruited by Tat to the 5'end of the nascent viral RNA. The authors propose that Tat might bind these mRNAs to drive a "feed forward" mechanism.

The Matrix protein (a subunit of Gag that is formed during virion maturation) was associated with mRNAs that exhibited a subset of enriched sequence motifs, potentially indicating an MA binding motif (although it should be noted that MA is not expressed in infected cells except as a subdomain of the Gag polyprotein).

Finally, seven of the top mRNA hits were knocked down in the SupT1 T cell line to see if they have relevance to infection. Knockdown of three targets (AFF2, RPLP0, and H4C9) was shown to yield reduced viral infectivity in a single round replication assay using a GFP reporter virus, suggesting functional relevance.

Overall, I liked the technical aspects of this paper very much. The approach seems robust and the use of multiple overlapping assays strengthened their pipeline. The paper is clearly written, and there is a lot of potentially useful information generated from the screen. On the other hand, I am less sure of the authors' functional studies. There are major caveats to overexpressing viral proteins independent of infection, and while it is intriguing that some of the mRNA hits may be relevant to HIV infection, these data are relatively preliminary with missing controls and insufficient validation.

Major comments:

1. I really appreciate the technical challenges intrinsic to this type of project. However, it needs to be better emphasized that during native infection viral proteins are expressed at defined amounts at defined times and operate in defined places within the cell. Because of this tight regulation, it can be easy to misinterpret results from overexpressing viral proteins at non-physiological levels that might be mis-aggregating, mis-trafficking, or at suboptimal stoichiometry relative to interaction partners. The tagged viral proteins from this study are from the prior Jäger et al. study that looked for protein interactors. This study yielded useful data- and this is a plus, but I think it still needs to be better stressed throughout (and especially in comparisons to prior studies that used virus) that the authors do not know if the protein levels are physiological or, more importantly, if the tagged viral proteins retain viral function. Related, I think the reader needs to be better reminded that the Gag subunits (notably MA and NC) are not generated until during/after budding, so that whatever is found bound to NC or MA alone in the cytoplasm is not really relevant to infection.

2. I think that data presentation in Figures 3 and 4 could be greatly improved- text is impossible to read in Figures 3 and 4A and in Figure 3 many interactors are impossible to track due to the figure's density. I'm also not sure exactly what the authors want the reader to glean from the interaction map even if it had bigger text- these data might be more easily navigated in a table?

3. Regarding host dependency factors, I found it interesting but surprising that 3/7 mRNA interactors were potentially involved in viral replication. Is the hypothesis that these mRNAs are bound by viral proteins to protect them or promote their expression? Or is it just as likely that there is a bias to the system and these are just enriched mRNAs that play big general roles in the cell? Importantly, there are no validation experiments on these hits so that at this point I'm reluctant to support the authors' conclusion that these are host dependency factors, this is the weakest part of the manuscript. The authors would need to first confirm protein knockdown and that the effects are not due to loss of cell viability and, ideally, define what is being blocked (e.g., it would seem most likely that these factors regulate viral gene expression?). Results are relatively preliminary on these points.

Minor comments:

1. Discussion paragraph describing MA-tRNA binding- if I remember correctly the result in the Kutluay et al. paper showed that Gag's association with tRNAs was lost when MA was deleted, which is a different situation from what the authors say here regarding MA alone binding fewer tRNAs than full-length Gag.

2. Figure 3- Gp120 is the ectodomain of a viral glycoprotein- are the indicated RNAs bound extracellularly? Are the Envelope derivatives trafficked appropriately?

3. Figure S8- it is surprising that CCNT1 knockdown did not affect the virus, this would have been expected as it is a major known Tat co-factor (and I have seen this used as a control in other studies).

We would like to thank all the reviewers for their constructive comments. Please find our detailed answers below to all these comments and questions (in blue).

Referee #1:

The manuscript by Schynkel et al. reports on the interactome of HIV proteins and host RNAs by using native RNA immunoprecipitation and sequencing (nRIPseq). The study provides a novel comprehensive resource for the HIV community. This is a carefully executed study with appropriate controls. To enrich for true interactors, the authors performed four assays, two with Jurkat T cells and in a subset two with SupT1 cells, uninfected and infected. However, the authors performed only very limited follow-up on only three targets, mRNAs and did not provide comprehensive validation on those mRNAs. It would strengthen the results of the manuscript by considering the following points:

- Jurkat T cells are treated with Doxycycline in comparison to the SupT1 cells. Dox treatment has been shown to influence the transcriptome profile. Do the authors suspect differences within Assay A and B to C (D)?

Indeed doxycycline treatment has been shown to influence transcriptome profiles and could have influenced the differences observed between AB vs CD. We have made this more clear in the discussion section as a limitation of our study (lines 348-353). Furthermore, pulldown strategies are very challenging protocols and a lot of experimental factors can contribute to the found interactors. It is for these reasons we have used extensive controls (IgG and GFP background) and two types of antibody-based pulldown strategies to look at the intersect for enriching valuable RNA interactors.

- Fig 1C and D: Why is the outcome of assay A to B so different

Although it seems like the outcome of Assay A to B is very different in Figure 1C and D, there was a significant overlap in number of interactors identified in both these assays for 5 out-of-6 HIV-1 proteins (Supplemental Table S3). Especially for NC and MA there was great concordance between these two assays. For the other HIV-1 proteins we indeed found a more limited overlap (yet significant, except for TAT). This could be explained by masking of the RNA binding place of the protein and sterical hindrance by the alternative pulldown method together with a more promiscuous binding of the FLAG antibody. In addition, we also refer to the possible reasons highlighted in the answer to the previous comment.

- Fig 2A: The assay result of D could be very informative, as here, the cells were infected. The authors should provide the identity of all overlapping interactors (in a table form for easy reference)

We have provided the identity of all overlapping interactors between the 4 nRIPseq assays in an overview table. This is added as Supplemental Data RNA interactors (Excel file>OVERLAP_ASSAYS) for ease of use of the reader.

- Fig 2B: significance asterisks are hardly visible.

We have adapted this.

- Fig 2C: please name the identity of the detected interactors (top 10 of each HIV protein), within this figure or in an extra table

We have provided the identity of these top 10 of each protein in an overview table. This is added as Supplemental Data RNA interactors (Excel file>TOP10).

- Fig 3 and 4a: please provide better resolution as text is blurry when zooming in

We have adapted this.

- Fig. 4c: please provide a table with the top 365 hits (identity) and respective overlap with screens

We have provided the identity of these top interactors and included the respective overlap with the screens mentioned by the reviewers (Brass, Konig, Yeung, Zhou, Park, Jager). This is added as Supplemental Data RNA interactors (Excel file>OVERLAP_SCREENING). After careful inspection, we have retrieved 373 interactors (instead of 365) that bind to at least one HIV-1 protein and have a mean interaction score >50. This was adapted in the manuscript.

- Fig 5: the authors should delineate whether the RNA is responsible for the effect they see or whether the protein that is coded by the RNA is responsible? The authors should measure the protein level (does it change?), are there inhibitors, activators for the proteins known that could be used for these experiments? What actually leads to HIV inhibition upon ASO treatment?

We have now carefully delineated in the manuscript that we cannot state if the loss of RNA expression (or protein expression) is responsible for the observed phenotype and that these types of experiments should be conducted in follow-up efforts. With our ASO-based screening experiments, we do demonstrate that if we target identified RNA interactors there is an effect on HIV-1 infection *in vitro*, highlighting the potential value of this first unprecedented host RNA - HIV protein interactome effort. We believe that more in depth and mechanistic exploration of the interactors lies outside the scope of this manuscript.

- The authors consider the screening hits as RNAs potentially acting as host dependency "factors", however the authors should consider that these RNAs could possibly also aid in HIV defense. Based on the GO groups there are some that might lead into this direction, "innate immune response", "ubiquitin-ligase activity"

We acknowledge that these interaction partners can also aid the host defence (restriction factor) next to being hijacked by HIV-1 (dependency factor). We have added this narrative in the manuscript.

- The term host dependency "factor" for RNAs seems quite odd..

Agreed, the term host factor refers typically to proteins but in a broader context can also be used for any cellular molecule such as an RNA molecule. We have rephrased this to RNA interactor as much as possible throughout the manuscript.

- The authors should compare their hits not only to the three mentioned siRNA screens, however also to the pooled shRNA screen (Yeung et al 2009) and CRISPR screens, e.g. Hiatt et al., Nat Comm 2022; Park et al., 2017; They could consider that the discovered RNAs might bind to a complex pulled-down by their approach, so possibly could bind in an indirect manner to host factors binding to HIV-proteins, and therefore might consider also proteomics studies such as Jager et al., 2011

We have performed these additional overlaps and added this in an overview table. This is added as Supplemental Data RNA interactors (Excel file>OVERLAP_SCREENING).

Referee #2:

The manuscript by Schynkel et al. provides a useful resource for knowing the RNA interacting partners of the HIV-1 polyproteins. To identify the RNA interacting partners, the authors have performed native RIPseq in overexpressed viral proteins. The findings from this study can be useful for other studies and yet the reviewer is wondering if the authors can consider looking into the following points:

- Have the authors validated some of the interactions using qPCR?

Per reviewers request we have performed qPCR validation on a selected group of RNA interactors that were relevant and for which enough leftover material was available. The group of RNA interactors we focused on was the TAT interacting RNAs identified in the Jurkat cells via FLAG-based pulldown. We have designed and used validated primer sets and normalized the expression of these RNAs to their expression in either the IgG or GFP control. Primer validation was performed on a 7-point standard dilution curve generated on gDNA retrieved from Jurkat cells, measured in triplicate. Three-out-of-five of these enriched interactors we could corroborate using qPCR comparing with both IgG and GFP control (AFF2, AFF3, RN7SL1) and one was only enriched when compared with the IgG control (RPL22). This was added as Supplementary Figure S8.

Figure for referee with unpublished data and its description has been removed upon request by the authors.

Figure. Fold Enrichment plots for the TAT-RNA interactors for the Jurkat cell lines with FLAG-based pulldown (Assay A). Expression is normalized to the IgG control.

- Could the authors explain a bit more about the differences in interaction in different cell lines? While the authors mention the masking of RNA binding site as a probable reason, which the reviewer strongly agrees with. But could there be other reasons? For both Assay A and C Flag-tag was used. While NP and Tat went down there was an enhancement in Matrix. Also, during virus infection of SupT1, NP and Tat show an increase, and probably some or all these RNA were identified in Assay A.

We have expanded the discussion on potential reasons for the differences in interaction in different cell lines in the manuscript. Although Jurkat and SupT1 cells are both cell lines derived from primary T cells, there are still inherent differences in the origin of these cells (T cell leukemia vs. T cell lymphoblastic lymphoma) making that there are differences in cellular steady state transcriptomes and phenotype of the cells (e.g. membrane markers). This influences the wiring of cellular processes such as antiviral defences (e.g. differences in interferon response) and makes that these cell models can behave differently to the expression of HIV-1 proteins or to HIV-1 infection. Hence, this is one of the reasons we included two cell lines and multiple background controls to zoom in on the intersection of interactors of these pulldown strategies.

Indeed, in the overlap diagrams in Figure 2A, there is an overlap between the interactors found in Assay A and Assay D for NP, TAT and MA. To further clarify all these overlapping interactors, we have added an overview table describing their identities in Supplemental Data RNA interactors (Excel File).

- Continuing on infection, could the authors look into what changes during infection between Assay C and D. More information on the types of RNA (coding or non-coding etc.) and the pathways affected? The biological processes have been focused more on non-infectious conditions.

In this rebuttal we have included several new overview tables with info on the discovered interactors (including biotype info), this as Supplemental Data RNA interactors (Excel File).

To specifically answer this comment, we have sub filtered on the interactors found for Assay C and D and placed this as an additional table in the Supplemental Data RNA interactors (Excel File>ASSAY_C_vs_D).

In terms of biological processes affected, we have performed DAVID pathway analysis for the interactors of each of the three HIV proteins (TAT:140 ,MA: 18, NC: 366) that were found only in assay D and not in assay C. We explored gene ontology (BP, CC, MF) and pathway databases (Biocarta, Reactome, KEGG) and used cutoff of 0.05 on the Bonferroni corrected p-value. Although limited enrichment was found, for MA, there was a significant enrichment of actin cytoskeleton and myosin complexes. This data is added as a Supplemental Table S7.

HIV-1 protein	Category	Term	Fold Enrichment	Bonferroni
TAT	GOTERM_MF_DIRECT	GO:0005515-protein binding	1.24826631	0.006328938
MA	GOTERM_CC_DIRECT	GO:0015629-actin cytoskeleton	39.31226054	0.000189992
MA	GOTERM_CC_DIRECT	GO:0016459-myosin complex	128.25625	0.011764858
MA	GOTERM_CC_DIRECT	GO:0005886-plasma membrane	3.042965709	0.108515049
MA	GOTERM_MF_DIRECT	GO:0003779-actin binding	40.39144385	1.98326E-09
MA	GOTERM_MF_DIRECT	GO:0051015-actin filament binding	39.9217759	9.75794E-05
NC	GOTERM_BP_DIRECT	GO:0043567-regulation of insulin-like growth factor receptor signaling pathway	54.08988764	0.038230388
NC	GOTERM_CC_DIRECT	GO:0070062-extracellular exosome	2.215420823	0.000120146
NC	GOTERM_CC_DIRECT	GO:0005829-cytosol	1.514344362	0.010804534
NC	GOTERM_CC_DIRECT	GO:0005739-mitochondrion	2.242123863	0.021270434
NC	KEGG_PATHWAY	hsa03010:Ribosome	5.661437908	0.033271568

Table. DAVID gene ontology and pathway enrichment results for the interactors of HIV-1 proteins (TAT, MA, NC) found in Assay D.

- In the infection model have the authors investigated interaction with viral RNAs?

As the focus of our manuscript is on human RNAs, we have not pursued this. Although we consider this as an interesting investigation, we have made all raw sequencing data of these RIPseq experiments (GEO) available for the scientific community to investigate this further.

- Most of the identified RNA were protein-coding genes (86%). How the interactions of the 3 major HIV proteins with these coding RNA could influence downstream signaling processes. The authors have investigated 15 RNA interactors, with further interrogation on H4C9, RPLP0, and AFF2. There were 19 RNA that overlapped with either Brass et al. or Zhou et al, and it is not clear whether these were used for screening. The reviewer is curious to know the details of these overlapping RNAs, maybe as a supplementary table.

To further clarify this we have added an overview table describing their identities. We have also included additional overlaps with other HIV-1 host factor screening efforts in Supplemental Data (Excel file>OVERLAP_SCREENING).

- In Fig5. For H4C9. Despite higher expression of ASO1, decreased infectivity. Have I missed something here?

Indeed, there was no knockdown present for H4C9 48h post infection for ASO1. However, there is a decreased infectivity seen 24 and 48h post infection. We hypothesize, as we do not have knockdown data 24h post infection, that there could have been a potential knockdown early on (e.g. at 24h post infection) that triggered the effect on infectivity but knockdown faded at 48h post infection. This in contrast to the decreasing effect on infection, as this takes longer to recover when infection is initially hampered, especially in a time frame of 48h. In addition, there is also the possibility that this particular ASO is triggering an off-target effect, explaining this phenotype. We have added this in the manuscript to make this observation more clear.

- Can the authors attempt knockdown in an HIV reactivation model like Jlat or U1 to observe the effects on reactivation?

Per reviewers request we have attempted knockdown in the J-Lat 10.6 HIV-1 reactivation model for 3 hits identified in our ASO screen: AFF2, H4C9, RPLP0. We have performed this with and without addition of SAHA. As a positive control to show reactivation potential of the J-Lat 10.6 we have used PMA.

There was limited knockdown observed for AFF1 ASO1 and good knockdown for H4C9 ASO1 (see below). However, despite the fact PMA reactivated J-Lat 10.6 to around 40%, we could not detect an effect on reactivation in the ASO-treated conditions that was different from the scrambled ASO control. This was added is Supplemental Figure S11 and S12.

Figure. Knockdown assessment via qPCR for the expression of AFF2, HC49 and RPLP0 in J-Lat 10.6 after treatment with ASO. Results are normalized to the scrambled ASO control.

Figure. J-Lat reactivation measured as GFP% cells via MACSquant flowcytometer in the context without (top panels) and with SAHA (bottom panels). PMA-treated vs ASO scrambled treated controls shown on the left, ASO-treated conditions shown on the right panels.

- Some Figures are difficult to read. In Fig2B. significant stars are difficult to read in the dark blue. Can consider changing them to white when it is a dark background. In Fig 4A font size should be increased considerably.

We have adapted this.

Referee #3:

Here, Schynkel et al. carry out native RNA immunoprecipitation and sequencing (nRIPseq) to identify host cellular RNAs associated with HIV-1 proteins/polyproteins purified from cell lysates. They identify several hundred RNAs that co-precipitated with 18 individual tandem-affinity-tagged viral proteins. Nucleocapsid, Rev, Pol, and Gag pulled down the most RNA interactors. These RNAs were mostly mRNAs but tRNAs, lncRNAs, pseudogenes were also ID'd. To enhance rigor, samples were collected from two independent cell types in four total assay configurations, allowing the authors to identify a subset of RNAs that were reproducibly IP'ed and, thus, their top binders.

Interestingly, Tat pulled down mRNAs encoding components of a transcriptional super elongation complex that included CCNT1 and 7SK RNA, known regulators of the pTEF-b protein complex that is recruited by Tat to the 5'end of the nascent viral RNA. The authors propose that Tat might bind these mRNAs to drive a "feed forward" mechanism.

The Matrix protein (a subunit of Gag that is formed during virion maturation) was associated with mRNAs that exhibited a subset of enriched sequence motifs, potentially indicating an MA binding motif (although it should be noted that MA is not expressed in infected cells except as a subdomain of the Gag polyprotein).

Finally, seven of the top mRNA hits were knocked down in the SupT1 T cell line to see if they have relevance to infection. Knockdown of three targets (AFF2, RPLP0, and H4C9) was shown to yield reduced viral infectivity in a single round replication assay using a GFP reporter virus, suggesting functional relevance.

Overall, I liked the technical aspects of this paper very much. The approach seems robust and the use of multiple overlapping assays strengthened their pipeline. The paper is clearly written, and there is a lot of potentially useful information generated from the screen. On the other hand, I am less sure of the authors' functional studies. There are major caveats to overexpressing viral proteins independent of infection, and while it is intriguing that some of the mRNA hits may be relevant to HIV infection, these data are relatively preliminary with missing controls and insufficient validation.

Major comments:

1. I really appreciate the technical challenges intrinsic to this type of project. However, it needs to be better emphasized that during native infection viral proteins are expressed at defined amounts at defined times and operate in defined places within the cell. Because of this tight regulation, it can be easy to misinterpret results from overexpressing viral proteins at non-physiological levels that might be mis-aggregating, mis-trafficking, or at suboptimal stoichiometry relative to interaction partners. The tagged viral proteins from this study are from the prior Jäger et al. study that looked for

protein interactors. This study yielded useful data- and this is a plus, but I think it still needs to be better stressed throughout (and especially in comparisons to prior studies that used virus) that the authors do not know if the protein levels are physiological or, more importantly, if the tagged viral proteins retain viral function. Related, I think the reader needs to be better reminded that the Gag subunits (notably MA and NC) are not generated until during/after budding, so that whatever is found bound to NC or MA alone in the cytoplasm is not really relevant to infection.

We agree completely with the reviewer and have elaborated on this more carefully and extensively in the discussion.

2. I think that data presentation in Figures 3 and 4 could be greatly improved- text is impossible to read in Figures 3 and 4A and in Figure 3 many interactors are impossible to track due to the figure's density. I'm also not sure exactly what the authors want the reader to glean from the interaction map even if it had bigger text- these data might be more easily navigated in a table?

We have changed this and added several table overviews to browse through the data. This is added as Supplemental Data (Excel file).

3. Regarding host dependency factors, I found it interesting but surprising that 3/7 mRNA interactors were potentially involved in viral replication. Is the hypothesis that these mRNAs are bound by viral proteins to protect them or promote their expression? Or is it just as likely that there is a bias to the system and these are just enriched mRNAs that play big general roles in the cell? Importantly, there are no validation experiments on these hits so that at this point I'm reluctant to support the authors' conclusion that these are host dependency factors, this is the weakest part of the manuscript. The authors would need to first confirm protein knockdown and that the effects are not due to loss of cell viability and, ideally, define what is being blocked (e.g., it would seem most likely that these factors regulate viral gene expression?). Results are relatively preliminary on these points.

We agree that the statement that these RNA interactors are host dependency factors is a bit bold and not fully supported by the experimental work presented in the current manuscript. Therefore, we have toned down our conclusions, emphasizing follow-up experimental work to confirm these potential factors and elucidate how these factors contribute to the HIV-host interplay.

Per reviewers request, we added the viability data of the ASO validation experiment, indicating a decrease for RPLP0 ASO2 at 24 and 28h post infection whereas for the other

conditions cell viability was within an acceptable range as compared with the scrambled ASO control. This was added as Supplementary Figure S10.

Figure. Cell viability data measured via PI staining (green) and HIV-1 infection levels via EGFP (blue) on MACSQuant flow cytometer for the ASO-treated conditions versus a scrambled-ASO control in SupT1 cells.

Minor comments:

1. Discussion paragraph describing MA-tRNA binding- if I remember correctly the result in the Kutluay et al. paper showed that Gag's association with tRNAs was lost when MA was deleted, which is a different situation from what the authors say here regarding MA alone binding fewer tRNAs than full-length Gag.

Indeed, when MA was deleted there was a large decrease in the capacity of tRNA binding of the Gag polyprotein albeit not completely. This would be suggestive of Gag-tRNA binding also being driven by other domains. Whether or not in our case we might be picking up more of these Gag-tRNA interactions that are non-MA driven, is largely speculative and requiring further dedicated experiments. Therefore, we argue that we sufficiently raised this discrepancy in our discussion section of the manuscript.

2. Figure 3- Gp120 is the ectodomain of a viral glycoprotein- are the indicated RNAs bound extracellularly? Are the Envelope derivatives trafficked appropriately?

We could indeed hypothesize that this can happen, to prove this we would need to perform GP120 membrane staining to see whether these can be appropriately trafficked to the cell membrane. With the scope of this manuscript in mind we prefer not to further speculate on this matter.

3. Figure S8- it is surprising that CCNT1 knockdown did not affect the virus, this would have been expected as it is a major known Tat co-factor (and I have seen this used as a control in other studies).

Indeed, for our ASO-based screen we have designed the ASO sequences ourselves according to an in-house developed design tool (available at <https://iomics.ugent.be/pjdev/design>). In our setup, we performed the ASO screen without prior validation whether these ASOs result in successful knockdowns. Only when we observed an effect on infection we assessed knockdown of the intended RNA target via qPCR. Our hypothesis is that these CCNT1 ASOs did not result in successful knockdown of CCNT1.

Dear Dr. Trypsteen

Thank you for the submission of your revised manuscript to our editorial offices. I have now received the reports from the two referees that I asked to re-evaluate the study, you will find below. As you will see, both referees now support the publication of the study in EMBO reports. Referees #3 has a remaining concern I ask you to address in a final revised manuscript. Please also provide a final p-b-p-response regarding this point of the referee.

- Please provide the abstract written in present tense throughout.
- Please provide individual production quality figure files as .eps, .tif, .jpg (one file per figure), of main figures and EV figures (see below). Please upload these as separate, individual files upon re-submission.

The Expanded View format, which will be displayed in the main HTML of the paper in a collapsible format, has replaced the Supplementary information. You can submit up to 6 images as Expanded View. Please follow the nomenclature Figure EV1, Figure EV2 etc. The figure legend for these should be included in the main manuscript document file in a section called Expanded View Figure Legends after the main Figure Legends section. Additional Supplementary material should be supplied as a single pdf file labeled Appendix. The Appendix should have page numbers and needs to include a table of content on the first page (with page numbers) and legends for all content. Please follow the nomenclature Appendix Figure Sx, Appendix Table Sx etc. throughout the text, and also label the figures and tables according to this nomenclature.

- We updated our journal's competing interests policy in January 2022 and request authors to consider both actual and perceived competing interests. Please review the policy <https://www.embopress.org/competing-interests> and update your competing interests if necessary. Please name this section 'Disclosure and Competing Interests Statement' and put it after the Acknowledgements section.
- Please reduce the number of keywords to 5 and order the manuscript sections like this, using these names: Abstract - Keywords - Introduction - Results - Discussion - Methods - Data availability section - Acknowledgements - Disclosure and Competing Interests Statement - References - Figure legends - Expanded View Figure legends
- We now use CRediT to specify the contributions of each author in the journal submission system. CRediT replaces the author contribution section. Please use the free text box to provide more detailed descriptions and do NOT provide your final manuscript text file with an author contributions section. See also our guide to authors: <https://www.embopress.org/page/journal/14693178/authorguide#authorshipguidelines>
- Please remove the section 'Materials & Correspondence' from the manuscript text file and move all the funding information to the Acknowledgements. Moreover, please make sure that all the funding information is also entered into the online submission system and that it is complete and similar to the one in the acknowledgement section of the manuscript text file.
- Please remove the referee token from the DAS (data availability section) and make sure that the dataset is public latest upon online publication of the manuscript.
- Please remove the list of abbreviations from the manuscript. Please define each abbreviation upon first mention in the manuscript text.
- Please make sure that the number "n" for how many independent experiments were performed, their nature (biological versus technical replicates), the bars and error bars (e.g. SEM, SD) and the test used to calculate p-values is indicated in the respective figure legends. Please also check that all the p-values are explained in the legend, and that these fit to those shown in the figure. Please provide statistical testing where applicable. Please avoid the phrase 'independent experiment', but clearly state if these were biological or technical replicates. Please also indicate (e.g. with n.s.) if testing was performed, but the differences are not significant. In case n=2, please show the data as separate datapoints without error bars and statistics. See also: <http://www.embopress.org/page/journal/14693178/authorguide#statisticalanalysis>

If $n < 5$, please show single datapoints for diagrams. It seems n is 2 for 5A/B. Thus, please show separate datapoints and remove the stats; or add a third replicate. Moreover:

- Please provide the exact p values in the legends of figures 2b; 5a-b (if a third replicate is added).
- Please indicate the statistical test used for data analysis in the legends of figures 2b; 4a; 5a-b (if a third replicate is added).
- Please note that in figure 2b; there is a mismatch between the annotated p values in the figure legend and the annotated p values in the figure file that should be corrected.

- Please add to each legend (main, EV and Appendix figures, where applicable) a 'Data Information' section explaining the statistics used or providing information regarding replicates and scales. See:

- Please make sure that all figure panels are called out separately and sequentially. Presently, there seems to be no callout for panel 4C. Please check.

- Please use our reference format:

- There are 2 zipped tables in Excel uploaded. I guess these are datasets. Please upload these separately as datasets using the nomenclature Dataset EV1, Dataset EV2 and add callouts to these using these names to the manuscript text.

In addition, I would need from you:

Best,

Referee #1:

The authors made a significant effort to address all issue raised. The publication is of interest to the community and well-suited for publication.

Referee #3:

This is a resubmission of a study from Schynkel et al. who apply a technique called nRIPseq to identify host RNAs that bind to individual HIV proteins that are inducibly expressed in multiple T cell lines. The authors have worked hard to address the bulk the issues brought up by reviewers.

My only significant remaining concern is the paper's discussion of Gag and Gag-Pol polyproteins. During HIV assembly MA, CA, NC, p6 Pol, RT, PR, and IN are typically not generated until assembly and protease activation. As such, I don't think that MA and NC interactions with host RNAs can be discussed as relevant to Gag function as they are in lines 271-276. Some qualifying language is added at the end of this paragraph but this aspect of the paper should still be improved.

Answers to the reviewer's comments

Referee #1:

The authors made a significant effort to address all issue raised. The publication is of interest to the community and well-suited for publication.

We thank the reviewer for acknowledging our efforts.

Referee #3

This is a resubmission of a study from Schynkel et al. who apply a technique called nRIPseq to identify host RNAs that bind to individual HIV proteins that are inducibly expressed in multiple T cell lines. The authors have worked hard to address the bulk the issues brought up by reviewers.

We thank the reviewer for acknowledging our efforts.

My only significant remaining concern is the paper's discussion of Gag and Gag-Pol polyproteins. During HIV assembly MA, CA, NC, p6 Pol, RT, PR, and IN are typically not generated until assembly and protease activation. As such, I don't think that MA and NC interactions with host RNAs can be discussed as relevant to Gag function as they are in lines 271-276. Some qualifying language is added at the end of this paragraph but this aspect of the paper should still be improved.

Indeed, we have added qualifying language to make this point clear. In addition, I have rephrased this section and subtitle, as the only point we want to highlight is that the majority of Gag interactors are shared with the NC interactors identified. This hints that the Gag-RNA binding could happen via its NC subdomain. I hope this makes our point clear that we do not transfer functions from MA/NC to Gag functioning but want to indicate potential NC subdomain driven binding of Gag with host RNAs.

Dr. Wim Trypsteen
Ghent University and Ghent University Hospital
HIV Cure Research Center, Department of Internal Medicine and Pediatrics
Corneel Heymanslaan 10
Gent, Oost-Vlaanderen 9000
Belgium

Dear Dr. Trypsteen,

I am very pleased to accept your manuscript for publication in the next available issue of EMBO reports. Thank you for your contribution to our journal.

Yours sincerely,
